# The Northumberland Exercise Referral Scheme as a Universal Community Weight Management Programme: A Mixed Methods Exploration of Outcomes, Expectations and Experiences across a Social Gradient

**DOI:** 10.3390/ijerph17155297

**Published:** 2020-07-23

**Authors:** Caroline J. Dodd-Reynolds, Dimitris Vallis, Adetayo Kasim, Nasima Akhter, Coral L. Hanson

**Affiliations:** 1Department of Sport and Exercise Sciences, Durham University, Durham DH1 3HN, UK; 2Wolfson Research Institute for Health and Wellbeing Physical Activity Special Interest Group, Durham University, Durham DH1 3HN, UK; a.s.kasim@durham.ac.uk (A.K.); nasima.akhter@durham.ac.uk (N.A.); 3Durham Research Methods Centre, Durham University, Durham DH1 3HN, UK; dimitris.vallis@durham.ac.uk; 4Department of Anthropology, Durham University, Durham DH1 3HN, UK; 5School of Health and Social Care, Edinburgh Napier University, Edinburgh EH11 4BN, UK; c.hanson@napier.ac.uk

**Keywords:** exercise referral, obesity, physical activity, inequalities, sociodemographic

## Abstract

Exercise referral schemes (ERS) are internationally recognised, yet little attention has been paid to discrete referral groups or the influence of wider social determinants of health. The primary quantitative element of this mixed methods study used a mixed effects linear model to examine associations of sociodemographic predictors, obesity class and profession of referrer on weight and physical activity (PA) variables for weight-related referrals (*n* = 3624) to an established 24-week ERS. Chained equations modelling imputed missing data. The embedded qualitative element (*n* = 7) used individual semi-structured interviews to explore participant weight-related expectations and experiences. Age, gender and profession of referrer influenced weight loss. PA increased and was influenced by age and gender. The weight gap between the most and least obese narrowed over time but the PA gap between most and least widened. Age, employment and obesity class were most predictive of missing data but would unlikely alter overall conclusions. Qualitative themes were weight-loss support, personal circumstances and strategies, and weight expectations versus wellbeing rewards. This ERS worked, did not widen existing obesity inequalities, but demonstrated evidence of PA inequalities for those living with deprivation. To improve equity of experience, we recommend further stakeholder dialogue around referral experience and ongoing support needs.

## 1. Introduction

Physical inactivity is a global pandemic [1] with an estimated annual worldwide cost burden of $53.8 billion [2]. Exercise referral schemes (ERS) are an internationally recognised means through which to prescribe physical activity (PA) [3,4], yet there remains limited understanding as to who such schemes might best work for, and why. In the UK, national ERS policy is broad [5], recommending primary-care-based referral as a means to promote behaviour change in sedentary or inactive patients who also present with another broadly defined health condition [5,6]. Schemes are usually 10–16 weeks [7] and led by qualified exercise professionals in the community, following a referral from a healthcare professional (HCP). Due to vague policy guidance, schemes are mostly universally implemented on the premise that facilitating access will translate to increased PA. We have previously criticised this approach, which fails to be sensitised to complex individual behavioural and social influences on PA participation [8]. Furthermore, systematic review evidence has raised questions about ERS effectiveness, due to heterogeneity of studies and cost effectiveness [9,10]. 

Clear understanding of ERS success is further complicated by the myriad health conditions with which individuals can be referred [5]. In a number of cases, ERS has inadvertently doubled as a weight management referral pathway for inactive overweight and obese patients [11,12]. This is problematic for PA referral pathways, which may not consider the established complex system of obesity and weight loss [13] in policy guidance, not least since ERS primary and secondary outcomes are usually based around PA change and scheme adherence, rather than weight change parameters [11,14,15,16]. Notwithstanding this issue, the impact on ERS effectiveness for other referral conditions has been examined previously with some success indicated for cardiovascular and mental health referrals [17]. For weight-related referrals, the evidence is scarce, however, and the true impact of referring patients with elevated weight to ERS needs to be understood, in terms of both PA and weight change, and engagement and retention [12,18,19,20,21,22]. What is clear is that ERS does not work for all; “success” can be influenced by socioeconomic and demographic factors, and participant relationships with ERS can be problematic and heavily influenced by complex life circumstance [21,23]. These issues together represent important gaps in the evidence base. As such, further research is required to better elucidate who ERS can work for, with attention specifically paid to health referral condition and equity of scheme access and engagement.

Prevalence of excess weight is unevenly distributed, and socioeconomic inequalities are well documented [24,25]. Socially disadvantaged groups, characterised by lower income, lower education and residing in more disadvantaged areas, are predisposed to higher levels of adiposity in high-income countries [26,27]. Indeed, inequalities are widening [24], posing a risk of intervention-generated inequalities due to the wealth of weight-management interventions and strategies in existence. Ahern et al. (2016) [28] acknowledged a perception that primary care weight loss referrals best serve “middle class, middle-aged women”, for example. Gender bias (towards women) has been reported in weight management referrals [29,30] and we reported that HCPs struggle to raise the conversation of weight [31], particularly with males. Promisingly, however, a few international reviews suggest that, at a community weight-management intervention level, weight related inequalities are not widened and in some instances may actually be reduced [24,32,33]. Mixed evidence is apparent for these effects in the case of individually targeted interventions [24,33] and there is a lack of evidence at societal level. Notably, there has been limited study within the UK [24,34]. The seminal review by Marmot [35] in 2010 reported that, if measures are not taken to reduce obesity inequalities, then the cost burden will reach £5 billion per year by 2025 for England and Wales. Arguably, at the time of writing, reducing inequalities related to obesity is far more advanced as a field of research than PA-related inequalities. Recent global data have, however, demonstrated an association between activity inequality and obesity, as well as activity inequality and reduced PA, particularly for females [36].

Exercise referral has an important role in advancing this field. Policy makers, commissioners, referring HCPs and scheme delivers must consider the impact of ERS referral in terms of widening, maintaining or reducing inequalities. Within the ERS literature, very little attention has been paid to weight and PA changes, and how these might be influenced by wider social determinants of health. To our knowledge, no study has yet explored equity of a largescale ERS in terms of effectiveness across a social gradient. To address this significant gap in the evidence base, we undertook a largescale mixed methods study of over 3000 weight loss referrals made to an established and rigorously designed [6,37] 24-week community-based ERS, based in Northumberland in the northeast of England. Northumberland is a region with pockets of particularly high and mixed deprivation. Unemployment is higher than the national average [38], and excess weight is also higher than the national average [39].

Study objectives were twofold. Firstly, to explore equality of ERS experience, we examined the impact on weight status and PA, specifically accounting for social determinants of health (age, gender, index of multiple deprivation (IMD) quintile and employment status), along with profession of referrer and World Health Organisation (WHO) obesity category. Secondly, to better understand ERS experiences for this group, we explored participant expectations and experiences relating to weight and PA through interviews conducted at the start and mid-to-end point of the scheme.

## 2. Materials and Methods

The study employed an embedded mixed methods design [40], where the qualitative dataset provided a supportive secondary role to the quantitative dataset. It encompassed a main, primary quantitative phase of data collection and a subsequent qualitative phase, implemented partway through quantitative data collection.

### 2.1. Setting and Design

The Northumberland ERS operated across nine leisure sites in northeast England. It was designated nationally as an “emerging practice” PA scheme by Public Health England [37] and provided access for patients with a variety of health conditions, including overweight and obesity, cardiovascular disease, mental health problems, metabolic disease and musculoskeletal, respiratory and neurological conditions. The 24-week scheme has been described in full previously [12], but, in brief, referrals were made by primary or secondary HCPs, with referral reason indicated on a standardised form. Participants accessing the scheme were encouraged to attend two supervised exercise sessions per week up to a maximum of 48 sessions. To the best of our knowledge, this is the first study to consider discrete referrals specifically made to ERS for weight status. We published ERS engagement experiences of a wider group of participants referred for a range of medical conditions who completed two semi-structured interviews [21], but here we focus on weight related experiences and conversations, which were not previously explored.

Guidance from NHS National Research Ethics Service confirmed that this quantitative element of this study was considered a service audit of anonymised data. Ethical approval was granted for the quantitative element by the Department of Sport and Exercise Sciences Ethics Committee at Durham University (Project SPORT-2019-03-11T111443) and for the qualitative element by Northumbria University Faculty of Health and Life Sciences Ethics Committee (Ref: 15-03-131781). 

### 2.2. Quantitative Sample

Anonymised data for all those referred to the Northumberland ERS between June 2009 and March 2014 with a primary reason for referral indicated as overweight or obesity (BMI > 24.9 kg/m^2^) were included in the quantitative element of the study (*n* = 3624). All those referred during this period with any other primary reason for referral were excluded from the study. Some predictive personal and referral characteristics have been previously reported for an initial wave of participants (*n* = 2233) who started the scheme between 2009 and 2010, but this included all referral conditions [12].

### 2.3. Quantitative Procedures

All reported measurements were taken by scheme staff. Age, gender, postcode, employment status, primary and secondary reason for referral and profession of referrer were recorded by scheme staff directly from the referral form. All data were entered into a Microsoft Access database by two trained leisure trust employees, independent of staff collecting measurements. Anonymised data were extracted and provided to the study team.

Baseline consultations took place at the leisure centres and involved a discussion about reason for referral, activity preferences, current activity and potential barriers to increasing activity. Weight was measured with shoes and bulky clothing removed and to the nearest 0.1 kg using Seca 761 scales. Height was measured without shoes to the nearest 0.1 cm using a portable Leicester stadiometer. Participants were instructed to stand as tall as they could before measurements were taken. Waist circumference (w/c) was measured to the nearest 0.1 cm against bare skin at the point of the participants’ natural waist using a fabric anthropological measuring tape. Anthropometric measures were repeated at 12 and 24-week consultations. Where possible, the same member of staff completed measurements at each time point. The Godin Leisure Time Exercise Questionnaire (GLTEQ) [41] was used to self-report PA, as recommended at the time by the British Heart Foundation National Centre toolkit for ERS [6]. Whilst a self-report measure, the GLTEQ has been validated extensively [42], and we applied a health contribution score during the analyses, as recommended more recently by Godin [43,44]. Participants completed this during baseline and 24-week consultations and via postal follow-up at 52 weeks. Three questions were asked regarding usual weekly frequencies for leisure time activity in the domains of strenuous, moderate and mild activity.

### 2.4. Qualitative Sample

We used early convenience sampling of referrals made with a primary referral reason of overweight or obesity during May–June 2013 to two of the nine leisure centres. All were eligible to take part in two semi-structured interviews (*n* = 42). Referrals to these centres included a broad adult age range, males and females and a range of economic circumstances. Later recruitment used purposive, heterogeneity sampling [45] to target those <50 years old.

### 2.5. Qualitative Procedures

Eligible participants were invited to take part in the study during telephone contact made by ERS staff. CLH, a Ph.D. student, conducted all interviews. Participants were informed that the investigator was a Ph.D. researcher and an employee of the organisation providing the ERS, that she was not involved in scheme delivery and that the aim of the study was to improve the ERS. Interested participants received postal information and returned signed consent to register. Interviews (prior to participants commencing the scheme and after 12-20 weeks) took place at the leisure centre, in private, at a time convenient for participants. Topics discussed in the baseline interview included PA history, experiences of and motivators for referral, perceptions and expectations of the ERS and perceived barriers and facilitators to taking part. Topics discussed in the second interview included experiences of the ERS, and barriers and facilitators to participation. One pilot interview was conducted with no changes made to the interview guide. CLH made detailed field notes that focused on participants’ social context, quality of interactions and researcher bias, and kept a reflective diary. Interview times ranged 43–62 min (median 51 min).

### 2.6. Quantitative Analyses

Anonymised data for those 3624 participants with a primary referral reason recorded as weight were imported in Stata Statistical Software 15 (StataCorp, Texas, TX, USA) for statistical analyses. A further variable was created to stratify BMI data according to WHO classification (overweight or pre-obese 25–29 kg/m^2^; obese class I 30–34.9 kg/m^2^; obese class II 35–39.9 kg/m^2^; and obese class III 40+ kg/m^2^) [46]. Weight, BMI, w/c, and Godin data were checked and cleaned. GLTEQ domains of strenuous, moderate and mild activity were multiplied by 9.0, 5.0 and 3.0 METS, respectively, and total score was calculated by summing domain scores. Values >119 were removed [42]. To extrapolate arbitrary scores to health risk, an additional Health Contribution Score (HCS) variable was calculated from moderate and strenuous Godin data [43,44,47], with an upper limit of 98 [42]. A score of <14 units was classed as insufficiently active (less substantial or low health benefits), 14–23 units as moderately active (some benefits) and ≥24 units as active (substantial benefits) [42,44,47].

To examine the association of the outcomes and relevant predictors, a mixed effects linear model was employed, analysing repeated measures over time by accounting for individual random intercepts and temporal random slope variation. Weight, BMI, w/c, Godin score and HCS were available as continuous data. For each of these outcomes, the regression model included potential confounders and their interactions with time points. Variables and interactions that were not supported by likelihood ratio tests statistics [48] were excluded from the models. However, variables that were directly related to the research questions or a priori identified as important confounders were retained in the models whether they were significant or not. The model building process adopted suggestions by Hosmer (2013) [49]. Multiple imputation was considered to investigate the impact of missing data on the results. A chained equations model was used to generate ten imputed datasets. Rubin’s rules were implemented to obtain pooled results from the ten imputed models. To accommodate correlated variables in the imputation models, augmented regressions were also considered [50]. The imputed data for each of the five models were generated using all explanatory variables and the dependent variable at each case. Logistic, multinomial and linear models were used for binary, categorical and continuous variables, respectively, in the chained equations regressions. Auxiliary variables were not included in the final imputations as their inclusion did not lead to any improvements.

Due to the censored nature of Godin scores, which implies values limited to a range (0–119) [42] and a right-skewed distribution, imputation by linear regression and the assumptions of normality that accompany it would generate data surpassing the range dictated by Godin score calculation, thereby producing impossible values that would not fit well with observed data [51]. To combat these issues, Predictive mean matching was used (with k = 5 nearest neighbours) for Godin and Godin HCS variables within the chained equations structure, providing an imputation framework that is less susceptible to model misspecification [52,53].

### 2.7. Qualitative Analyses

Interviews took place between May and December 2013, were audio recorded and transcribed verbatim. All data including references to weight were extracted into a Microsoft Excel (Microsoft Corporation, Bellevue, Washington, WA, USA) spreadsheet. Pseudonyms were used and all data were thematically analysed using the framework approach [54]. CLH and CDR familiarised themselves with transcripts through several readings and CLH checked accuracy against audio recordings. We separately used line by line coding to openly record preliminary concepts and patterns for both interviews for two participants and ordered these using Microsoft Excel. This resulted in 18 initial codes including personal perceptions of weight, perceptions of current diet, personal support for weight loss, weight loss expectations, other expectations from scheme participation, age considerations, benefits of participation and support from primary care. After discussion, we established agreed codes to form an initial analytical framework of four themes: support in primary care, personal circumstances, expectations versus rewards and wellbeing strategies. We analysed one more set of transcripts, before refining and finalising of the analysis framework, condensing the initial framework to three final themes. We mapped and explored connections within and between participants and categories using a matrix. Our analysis developed, reviewed and finalised themes that identified the complexity of, and possible explanations for, the ERS experiences of those referred for weight loss.

## 3. Results

### 3.1. Quantitative Findings

Between June 2009 and March 2014, the ERS received 9002 referrals. Of these, 3624 (40.3%) had a primary referral reason recorded as weight (BMI > 24.9 kg/m^2^), accounting for the greatest proportion of all referrals. After checking and cleaning, the final database consisted of 3600 referrals. Of these, 2724 (75.7%) started the scheme, 1397 (38.8%) continued to 12 weeks and 1061 (29.5%) completed the 24-week scheme. The majority (69.4%) of weight-based referrals were female (Table 1), with a mean age of 51.3 years (SD 15.6). Overall, mean age at point of referral was 49.1 (SD 15.5) and at point of completion 56.3 (SD 13.7).

Table 2 presents available data (weight and PA) across time points, for those who completed the scheme.

Compared to the start of the scheme (but holding all variables and interactions constant at baseline), referrals had significant weight loss (−1.43 kg and −1.72 kg at 12 and 24 weeks, respectively, *p* < 0.01). Similarly, w/c (−2.14 cm and −2.57 cm, *p* < 0.01) and BMI (−0.51 kg/m^2^, *p* < 0.01, and −0.67 kg/m^2^) declined over the same periods. Godin PA scores rose by 16.30 at 24 weeks (*p* < 0.01). Holding variables constant at baseline, the increase only persisted by 4.33 compared to baseline, at 52-week follow-up (non-significant) and, similarly, the Godin HCS rose by 13.96 (*p* < 0.01) and 4.88 (*p* < 0.05) at 24 and 52 weeks, respectively, compared to baseline (Table 3). Interactions are explained below and provide a more complete representation of changes over time, which actually are more pronounced at 52 weeks than implied by the time effect alone. In terms of an illustration of contribution to health (from moderate and strenuous activity only), mean (SD) Godin HCS values rose from baseline 6.99 (13.35) to 19.65 (18.33) and 16.73 (17.73) at 24 and 52 weeks, respectively, falling short of the 24-point cut-off for “sufficiently active” [44] but indicating levels of moderate activity and thus associated health benefits [42,44]. It should be noted that these mean data reflect the score at those timepoints only, and do not account for influence of other variables (unlike the full prediction model).

A reduction in weight was apparent with increasing age and weight and w/c were higher among males than females. IMD quintiles held no associations with weight, BMI or w/c (Figure 1).

Considering profession of referrer, participants referred by practice nurses had significantly lower BMI and w/c than those referred by a GP, yet referrals made by others (or not stated on the form) indicated higher weight and w/c values (Table 3). Interaction effects were found for time × obese class for weight and BMI. At 12 weeks (Figure 2), weight and BMI reduction was greatest for obese classes II and III compared to pre-obese and this interaction persisted at 24 weeks for obese class III. More specifically, the magnitude of the difference in weight between those classed as obese Class III at point of referral and those classed as pre-obese at point of referral began to reduce at 12 weeks (−1.59 kg, *p* = 0.000), and the effect was dampened but persisted at 24 weeks (−2.32 kg, *p* = 0.000). In other words, weight loss was relatively greatest for those in the higher BMI category with a diminishing effect as time went on. The same pattern was true for BMI at 12 weeks (−0.63 kg/ m^2^, *p* = 0.000) and 24 weeks (−0.94 kg/ m^2^, *p* = 0.000) (Table 3).

Physical activity increased significantly between referral point and 24 weeks (Godin score and Godin HCS) and was reduced at 52 weeks but remained higher compared to baseline for Godin HCS. PA (Godin and Godin HCS scores) was negatively associated with age and was higher for males compared to females. In terms of interactions, at 24 weeks, PA was generally lower (Figure 3) for those residing in areas of greater deprivation, compared to those least deprived (for Godin/Godin HCS, −6.3/−3.4 for Quintile 1, −5.6/−2.9 for Quintile 2 and −4.9/−3.9 for Quintile 4). By 52 weeks, this effect largely disappeared.

For Godin PA, there was an interaction for both obese class II and obese class III at 52 weeks, compared to pre-obese. Here, PA increase was greater for these groups compared to pre-obese. The interaction held for Godin HSC, but only for obese class III. (Figure 4), showing that this group was more likely to maintain PA than those pre-obese.

#### Missing Data

Missingness is largely associated with dropping out of a scheme, which in turn implies higher amounts of missing data for each additional time point. Any further missing data may be a result of failure to collect or to report information. Patterns for missing data are shown in Appendix A.

There is variation in the patterns of missing data between variables, with Godin exhibiting much higher missingness for the third period. In general, all model variables were predictive of missing data to some degree, with the most predictive among them being age, employment status and obesity class. Older referrals and those who were retired implied a lower probability of missingness, while higher obesity class implied a higher likelihood of missing data.

The imputed models for Weight, BMI and w/c were very similar to the original models. There were a few exceptions where conclusions from original data and multiple imputations are different. Even in those exceptions, however, the sign and magnitude were similar. The Godin and Godin HCS imputation models were also similar to the original models albeit slightly less, with estimate magnitudes and significance varying for the IMD quintile variable and its interaction with time. These results indicate that missing data were not likely to significantly change the interpretation of results and associated conclusions.

### 3.2. Qualitative Findings

Seven participants, three recruited via convenience sampling and four via purposive, heterogeneity sampling, all referred from primary care, took part in initial interviews and five in second interviews (Table 4). The two participants who did not attend second interviews had brief telephone conversations with CLH approximately 12 weeks after their initial interview. Both had dropped out of the ERS. Louise had a new job and Vicky had new family caring commitments due to illness. Three weight-related themes developed: (1) support for weight loss in primary care; (2) personal circumstances and weight loss strategies; and (3) weight expectations versus wellbeing rewards.

#### 3.2.1. Support for Weight Loss in Primary Care

Those participants who raised the issue of weight with HCPs expected to receive dietary advice via a referral to a dietician or some written advice *“a type of diet sheet or something”* (Julie, first interview). Instead, HCPs presented the ERS as a weight loss programme:
“I had already asked to be referred to a dietician but I never got a letter through for it... so I never got the referral... so she signed us up to this programme and that was basically it... what I could be offered in terms of weight loss...”(Amy, first interview)

Some participants were already receiving weight loss support. Paul and Jackie had received dietetic support, Louise diabetic nursing support and Vicky ongoing GP support. For them, HCPs presented the ERS as an aid to weight loss: *“The doctor has put me on Orlistat as well, and this scheme will give me more exercise a week”* (Louise, first interview).

During her second interview, Amy suggested the need for GPs to be “*more proactive when your weight is lower*” stating “*it’s not just down to a diet plan, it’s down to… lifestyle change*”. (Amy, second interview).

#### 3.2.2. Personal Circumstances and Weight Loss Strategies 

Participants perceived that being overweight had a negative effect on their lives. A lack of confidence related to body image prevented some from everyday activities (Amy, Jackie, Patricia) *“I can’t go out places when people go… oh you’re fat*” (Jackie, first interview). Participants attributed weight gain to several personal factors: increasing age (Julie, Louise and Paul), reduced PA (Julie and Paul) and having children (Amy and Louise). Social support for weight loss varied from being very positive *“my parents are behind me as well… I will get help if I need help to look after my daughter”* (Louise, first interview) to a lack of family support, as reported by Patricia when discussing Slimming World *“and my husband said to me... it’s a lot of money for somebody to tell you that you are fat. He says I will tell you it for nothing”* (Patricia, first interview).

During initial interviews, it was apparent that all participants had made previous attempts to manage their weight. Strategies included increasing PA (Louise and Julie) and use of commercial weight loss programmes (Weight Watchers and Slimming World) with success followed by weight regain (Patricia, Louise, Julie and Vicky).

“*Well I’ve done Weight Watchers and that but I still haven’t lost... Well I did lose weight but then I did what they tell you what you are going to do... Diet, lose weight and then put it back on again*”.(Julie, first interview)

Participants felt that the ERS was a positive step in their weight management journey “*I felt happy about finally being able to do something about my weight”* (Amy, first interview) and were looking forward to seeing tangible benefits:
“*Well I’m looking forward to being thinner hopefully and a better shape so therefore just feeling healthier in myself. Also if I lose the weight that should prolong the time before I have to [have a knee operation]*”.(Julie, first interview)

#### 3.2.3. Weight Expectations versus Wellbeing Rewards

All participants had high expectations of weight loss during ERS participation. Based on UK public health information that sustainable weight loss is achievable by losing 1–2 lbs (0.45–0.9 kg) per week, most (Julie, Amy, Louise, Vicky, Patricia and Jackie) stated that a weight loss goal of 2–2.5 stones (12.7–15.4 kg) was a realistic target over the 24 weeks:
“*At least two stone if not more… probably just the two stone because it is always hard work for me*”.(Patricia, first interview)

Paul was more ambitious (three stones, 19.1 kg), although he was unsure about the reality of this stating *“if I’ve lost half a pound every week or even every month it would give me more incentive”* (Paul, interview 1). He was not alone in worrying that he may not achieve his weight loss goal, as others (Julie, Amy, Louise, Patricia and Vicky) all stated that, as long as they could see some consistent weight loss, they would be happy:
“*As long as I am losing something and I am feeling fitter then I will know that I am doing the right thing*”.(Louise, first interview)

During second interviews, it was apparent that those who attended the ERS achieved modest weight loss (reporting approximately 7 lbs or 3.2 kg). Despite this, these participants reported other benefits from taking part, such as improvements in overall physical and mental health that mitigated the lack of weight loss:
“*I wouldn’t say so much with my weight but my overall health, it’s a lot better. I’ve lost half a stone so obviously I feel a bit down… not [disappointed] as much as I thought I would be because… I can actually get into my jeans and I have actually dropped a dress size as well…*”(Amy, second interview)

Second interviews took place part way through the ERS and there were ongoing expectations for continued weight loss. Goals were revised based on continuing to achieve weight loss at the same rate as that seen to date. When asked whether her original weight loss goal was still realistic, Patricia responded: *“No I don’t think so. I might lose a stone maybe… but as long as I lose something”* (Patricia, second interview).

## 4. Discussion

This study considered outcomes and associated experiences for overweight and obese patients referred to a public health commissioned ERS. To our knowledge, it is the first to explore equity of a largescale ERS in terms of effectiveness across a social gradient. Furthermore, it is the first to analyse data for those specifically referred because of weight status. Interviewees represented a wide social gradient (Table 4), five of seven residing in the most deprived areas, with a mix of employment status. A wide age-range was represented at interview (20–29 to 60–69 years), but only one male, perhaps not surprising given the lower uptake of the ERS for men. We report here the success of ERS for weight referrals, both for increased PA and modest weight lost, but with some caveats around equity of provision, participant experiences and follow-up as discussed below.

### 4.1. Physical Activity and Weight Loss

Exercise referral is not intended as a weight loss programme per se [5], yet our study found that participants did experience weight loss. Our findings are favourable compared to little or no weight loss reported for general referrals across a number of other ERS [9,22]. In total, 1061 participants completed the ERS and reported some level of increased PA, indicating that that the scheme is to a large extent successful in its intended outcome (PA change) for those who adhere. Full comparison to other studies is limited by many different PA questionnaires used, and differing durations of schemes, as well as no direct comparison for weight-related samples independently; however, in their systematic review including a range of referrals, the authors [9] reported increased PA which largely fell short of meeting PA guidelines. Our findings concur with this in terms of any need to achieve moderate PA, however we revisit this point later in the discussion, in respect of recently revised PA policy and guidance. The Welsh National scheme’s RCT [14] also reported increased PA across CHD and mental health referrals at 16 weeks, compared to control. Furthermore, data from the UK National Exercise Referral Scheme Database [55] highlighted potential benefit in schemes up to three months, but also suggested that a high proportion of self-reported PA in their study was classed as “moderate” before the ERS began. Notably, we accounted for drop-out through missingness analyses which indicated that the positive changes observed would have been similar had all participants continued with the scheme. This is important as it shows that the improvements in PA and weight are not biased towards those most engaged (i.e., those who did not drop out).

Considering the modest weight loss, it may be that people remained on the scheme for reasons other than weight loss alone, something which is supported by our qualitative findings about the importance of improved wellbeing. Qualitative findings suggested that initial expectations of weight loss were high, although weight loss experienced was moderate for the participants interviewed, similar to the modest weight loss reported in our quantitative findings. Despite this, most of the participants interviewed who had continued with the ERS (had second interviews) were philosophical and tended to describe success in terms of improvements in psychological and physical wellbeing, similar to the findings of Pentecost and Taket (2011) [56]. For others, it is possible that the moderate weight loss was ultimately not sufficient to motivate engagement and this may be one of the reasons for drop-out. Eynon et al. (2019) [57] supported to this, suggesting lower expectations for change tend to be associated with better ERS adherence. Swift et al. (2014) [58] suggested that health professionals should educate participants about reasonable weight-loss expectations, whilst emphasising the many other related health benefits of PA programmes. We suggest that adherence might be improved if such discussions were undertaken by HCPs at the point of referral and continued by ERS staff during the initial consultation. Given the recommendation by the National Institute for Health and Care Excellence [5] that research is required into the impact of ERS on health outcomes in specific populations, we consider this an important finding for weight referrals which will be of use to other ERS commissioners and research teams when designing, evaluating and scaling-up existing and future schemes.

### 4.2. Sustainability, Adherence and Support Needs

In terms of sustaining PA after the ERS, referrals reported some increase in PA was maintained at 52 weeks (28 weeks after the ERS had finished), however this was not the case for all referrals. Of specific note is that PA increase was maintained for those in the higher obese class, but not for those classed as overweight. This suggests that bespoke follow-up support may be required by certain groups. It is useful to further interpret overall PA data in terms of contribution to health. Although the mean increases in PA at 24 and 52 weeks did not place referrals into the “sufficiently active” category (whereby they would be meeting the American College of Sports Medicine’s moderate/vigorous PA recommendations) [42,43,44] based on HCS scores, they would have moved from “insufficiently active” to be “moderately active” at 24 and 52 weeks [42,43,44]. Recently revised PA guidelines acknowledge the health and quality of life benefits which can be achieved from small changes below recommended thresholds [59], and as such we demonstrate an important positive change. Only a few studies, and none only for weight referrals, have included a follow-up period similar to ours [3,14,60] and success has been mixed.

To encourage PA in the months following scheme end, embedded social support mechanisms have been described as contributing to success in a smaller-scale Spanish ERS [3] and included a variety of approaches through the session leader and exercise groups themselves. Notably, both social support and social structures have been previously reported as influential in weight management decision-making [61] and both were highlighted through the present study. Family support (or lack of) was perceived as either a barrier or facilitator to weight loss, suggesting that social support, particularly personal relationships, may be critical in ERS success. Participants explained their weight gain in terms of personal factors throughout their life trajectory, including age and having children, but described this retrospectively and also highlighted previously unsuccessful attempts to lose weight through other commercial weight loss schemes. Furthermore, in their systematic review of psychosocial factors associated with ERS adherence, Eynon et al. (2019) [57] reported that intrinsic motivation, psychological need satisfaction, social support and self-efficacy were all associated with ERS adherence. Buckley and colleagues (2018) [62] conducted a participatory co-development phase for an ERS and reported the need for a culture shift to provide a more holistic PA referral approach, through staff appropriately trained in behaviour change. Given the success for some of our interviewees, and indeed for 1/3 of all referrals explored in our study, we believe that there is potential to adapt the scheme for optimal weight loss across a social gradient, but suggest that in doing so there must be full consideration of social and psychological factors from the outset. We also recommend that post-scheme exit support needs to be considered by commissioners and deliverers when designing ERS, but should be specifically targeted to the needs of different groups.

### 4.3. Inequalities at Referral Point

Overall, weight status tended to be greater for younger referrals (different to current population trends [63]) and for males. The mean age of weight-related referrals to the Northumberland ERS being 49.1 years and women comprising ~2/3 of all weight-related referrals (and completers) suggests that HCPs should, for equity, also consider targeting younger patients for ERS if weight loss is the primary reason for referral. They should also explore ways to raise conversations of weight and PA with males, in accordance with a similar message we have reported recently [31]. Exploration of these factors is required in the healthcare referral setting to ensure equity of referral. For example, it could be that some HCPs are less confident in recognising lower levels of excess weight, or that men tend to visit the GP Surgery less than women, or that the short window available for appointments is insufficient to raise issues of weight or inactivity in certain cases. Qualitative findings support the need for changes required at the point of referral. When meeting with HCPs, participants described an initial a perception that weight-loss support would be primary-care-based, participants considering that (for example) a dietitian would be the appropriate referral route, or that the exercise scheme would be a referral supplementary to something else (weight-loss medication or other support). This notion of patients preferring one-to-one personalised support, or receiving similar as part of a multidisciplinary approach has been reported elsewhere [64] and may similarly be a reason for lack of engagement or adherence in weight-loss referrals made to ERS.

### 4.4. Impact of ERS on Existing Inequalities

In terms of socioeconomic position, for weight loss, this largescale ERS proved equitable across a social gradient. This is an important finding for a weight referral-based scheme, particularly when existing comparable evidence (which suggests similar), consists of wider “whole-of-community” interventions [32]. Furthermore, Hillier-Brown et al.’s [34] systematic review reported the majority of intervention success with targeted interventions, rather than across a social gradient (such as in the present study), and even then, only in the short-term (~12 weeks). We demonstrate here the specific potential for ERS to reduce obesity inequalities by narrowing the “gap” between those most and least obese.

Higher levels of PA reported by younger referrals and males is similar to epidemiological trends, [65] although for males is perhaps counterintuitive considering they had greater overall weight status in our study. Existing age and gender-based PA inequalities were therefore not exacerbated by this ERS. By the end of the ERS, there was, however, evidence of intervention-generated inequalities, based on deprivation. This demonstrates the need for caution in development and implementation of community-based ERS that are not targeted to particular SES groups. We suggest that those living with greater deprivation may engage with ERS PA less well and recommend that this potential effect is noted at the point of ERS design by commissioners and researchers. It may well be that some targeting of ERS support is required for some lower SES groups. Of note, however, is that the PA inequality did not persist at follow-up. Whilst this is positive, we note a likelihood of reduced compliance with PA data reporting over time; only half of the 24-week sample was available for analysis at 52 weeks, potentially due to the postal return (rather than completion with scheme staff at 0 and 24 weeks).

Although our analyses account for missing data, we should note that only ~30% of all overweight/obese referrals actually completed the 24-week scheme. The rate of completion is lower to that observed across the scheme for generalised referral types (43%) [12] and lower than reported in other schemes of varying duration (43.8%) [14] (39%) [66]. Between referral point and scheme completion, average age shifted from 49 to 56 years old and the proportion of those classed as retired was substantially higher, similar to previous findings [12]. Reasons for low retention have been quantitatively explored across a range of referral reasons and studies suggest discrete factors including smoking and presence of moderate-to-high co-morbidities [11]. However more multifaceted reasons for non-engagement have been highlighted by Morgan et al. (2016) [19] through systematic appraisal of qualitative literature suggesting again that support mechanisms, and individualised, personalised provision, along with scheme setting, may be important in promoting ERS retention. Our qualitative findings concur to a large extent with multifaceted reasons for problematic engagement and lend a health equity lens to these complex factors. We suggest that ERS engagement may be even more complex for certain groups.

Based on our robust quantitative analyses, we suggest that the Northumberland ERS is operating well as a universal intervention for modest weight loss. The next step is to consider adapting the scale and intensity of the intervention, across the social gradient and proportionate to the level of disadvantage, particularly in respect of PA [27,35]. The exact nature of this should be considered by all stakeholders including commissioners, HCPs, deliverers and naturally, referrals themselves, but might incorporate elements of co-production with those key people [62]. Whilst further inequalities (as described for quantitative data) were not explicitly highlighted in developing themes, more subtle references were made to point to inequalities of social support and personal circumstances. Our qualitative findings thus provide insight into how managing initial expectations of weight loss, personal circumstances and social support needs are important. They suggest generalisability in a naturalistic sense [67] and will be important for policy makers, clinicians and other stakeholders to consider where ERS has weight-related referrals.

### 4.5. Strengths and Limitations

We would note that our study was not an RCT, rather it represented the naturalistic setting that community-based public health interventions serve. The study relied on measures taken by others and as such we cannot guarantee the accuracy of objective measures. We strongly believe, however, that studies such as this, which take place in the naturalistic environment, are essential in informing public health and clinical practice. Naturalistic studies provide a clear window through which to examine outcomes and reasons for those outcomes, particularly given the numerous and wide-ranging ERS currently operating internationally.

The service provider administered the GLTEQ according to the recommendation of the British Heart Foundation National Centre toolkit; however, it is not without limitation. Of course, use of objective monitoring is preferable for most accurate representation of activity; however, for largescale naturalistic public health evaluation, this is not feasible. Moreover, The GLTEQ has been extensively utilised and validated [44]. Typically, the questionnaire provides a score which has been associated with changes in aerobic fitness [41] but cannot be linked directly to PA recommendations. To provide a more meaningful interpretation, however, for this paper, we recalculated PA scores based on moderate and strenuous activity [43]. As such, the new values better reflect moderate-to-vigorous PA recommendations as understood internationally, and associated health benefits.

We used early convenience sampling to recruit participants due to ERS staff facilitating recruitment, and note previous criticism of this strategy [68,69]. However, we combined this with purposive sampling after the first three participants, to include those under 50 years.

Missing data due to drop-out are usual in public health programs and are certainly to be expected in ERS [12]. A strength of this study is the imputation to account for missing data, which demonstrated imputed models similar to originals. This tells us that findings from our study would have been similar if all referrals who began the ERS had remained on the scheme and thus strengthens the statistical and naturalistic generalisability of messages in this paper.

## 5. Conclusions

This study demonstrated success in increased PA and moderate weight loss for weight-loss referrals made to a largescale, nationally-recognised ERS. For those working with public health or clinical commissioning for weight-loss, we recommend that ERS has potential to reduce obesity inequalities, but care should be taken to ensure provision (ERS access, potential for PA change and support) is equitable across a social gradient. In terms of future directions for ERS, this might be best served through creation and adoption of clear policy around the referral process (setting of realistic health-related expectations) and via ongoing stakeholder dialogue around social support to aid PA engagement during, and following, ERS.

## Figures and Tables

**Figure 1 ijerph-17-05297-f001:**
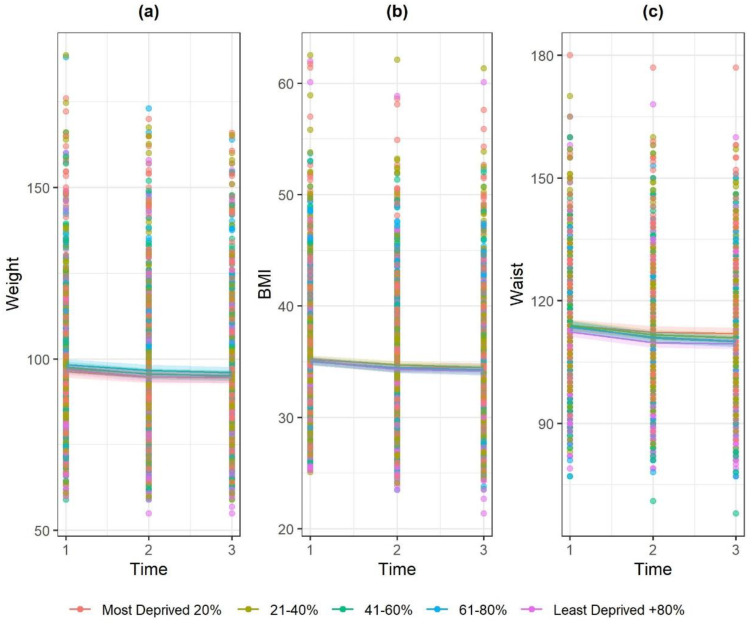
Marginal effect interaction plots of timepoint × IMD quintile for (**a**) weight, (**b**) BMI and (**c**) waist circumference.

**Figure 2 ijerph-17-05297-f002:**
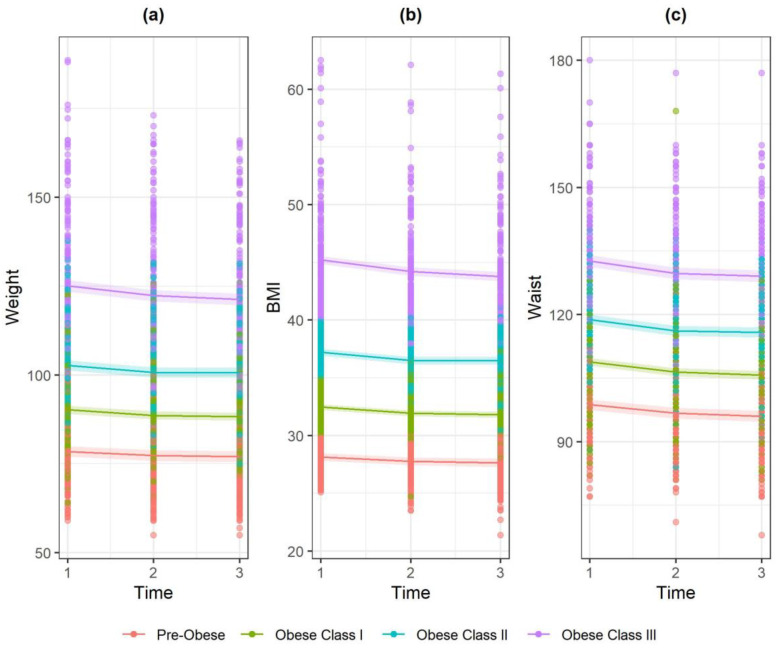
Marginal effect interaction plots of timepoint × obese class for: (**a**) weight; (**b**) BMI; and (**c**) w/c.

**Figure 3 ijerph-17-05297-f003:**
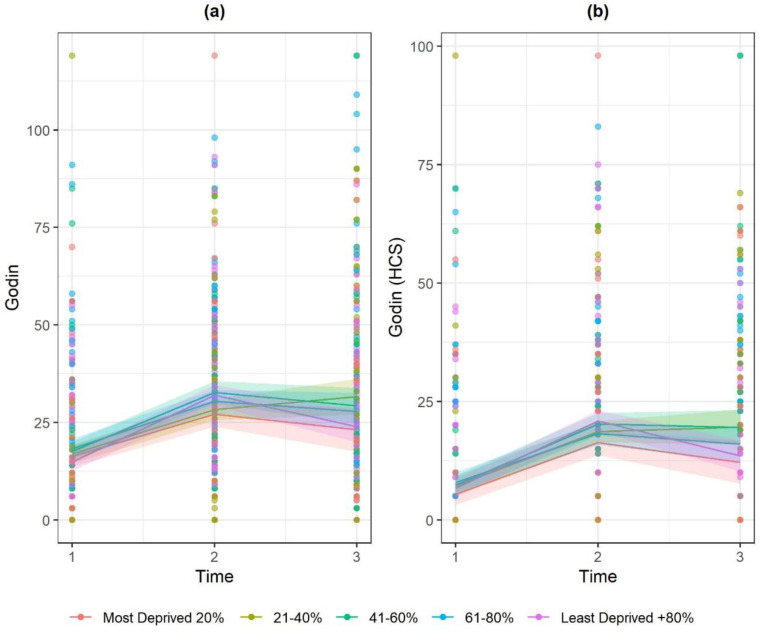
Marginal effect interaction plots of timepoint × IMD for: (**a**) Godin; and (**b**) Godin HCS.

**Figure 4 ijerph-17-05297-f004:**
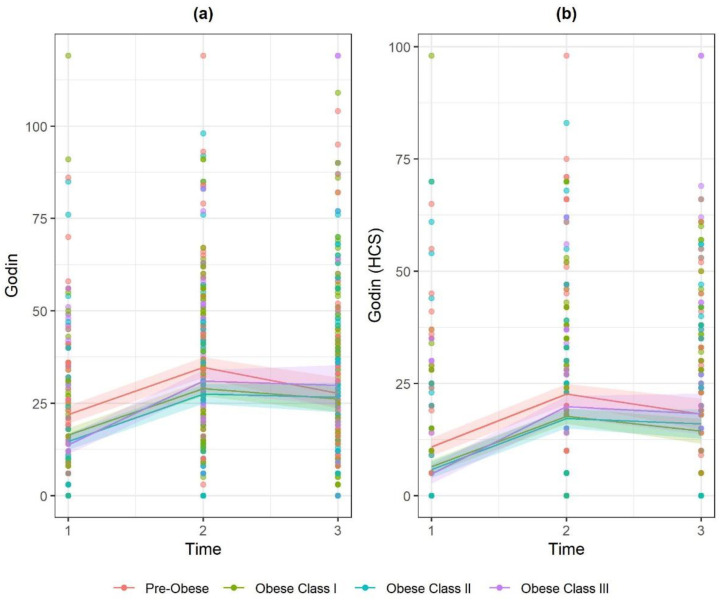
Marginal effect interaction plots of timepoint × BMI Class for: (**a**) Godin; and (**b**) Godin HCS.

**Table 1 ijerph-17-05297-t001:** Sociodemographic characteristics of overweight and obese referrals.

Variable	At Point of Referral *		At Point of Completion	
	**Frequency**	**%**	**Frequency**	**%**
**Gender**	**3585**	**100**	**1056**	**100**
Male	1098	30.6	351	33.2
Female	2487	69.4	705	66.8
**IMD Quintile**	**3595**	**100**	**1060**	**100**
20% Most Deprived	766	21.3	182	17.2
21–40%	852	23.7	237	22.4
41–60%	709	19.7	211	19.9
61–80%	574	16.0	188	17.7
81–100% Least Deprived	694	19.3	242	22.8
**Employment Status**	**3291**	**100**	**999**	**100**
Retired	866	26.3	455	45.6
Claiming Benefits	549	16.7	108	10.8
Employed or FTE	1183	36.0	246	24.6
Other/not Stated	693	21.1	190	19
**Profession of Referrer**	**3576**	**100**	**1058**	**100**
GP	2340	65.4	653	61.7
Practice Nurse	983	27.5	326	30.8
Other/not Stated	253	7.1	79	7.5
**BMI Category**	**2687**	**100**	**1045**	**100**
Pre-obese (25–29.9)	451	16.8	217	20.8
Obese class I (30–34.9)	894	33.3	386	36.9
Obese class II (35–39.9)	685	25.5	244	23.4
Obese class III (40+)	657	24.5	198	19

Bold text indicates total values. * Includes non-completers.

**Table 2 ijerph-17-05297-t002:** Number of observations available for each outcome across time points.

Variable	Observations at Baseline	Observations at 12 Weeks	Observations at 24 Weeks
Weight	1046	945	1052
BMI	1045	940	1050
Waist	1038	940	1044
	**Observations at Baseline**	**Observations at 24 Weeks**	**Observations at 52 Weeks**
Godin	1020	950	437
Godin (HCS)	1023	955	462

Observations represent scheme completers-only and exclude cases with missing data.

**Table 3 ijerph-17-05297-t003:** Factors associated with longitudinal change in weight, BMI, waist circumference, Godin and Godin HCS, based on mixed effects analysis.

Main Effects	Weight	BMI	Waist	Godin	Godin (HCS)
12 Weeks (Godin: 24 Weeks)	−1.429 (−2.083; −0.776)	−0.513 (−0.756; −0.271)	−2.138 (−3.042; −1.234)	16.299 (12.205; 20.392)	13.963 (10.545; 17.381)
24 Weeks (Godin: 52 Weeks)	−1.721 (−2.518; −0.924)	−0.673 (−1.005; −0.341)	−2.567 (−3.518; −1.616)	4.325 (−1.054; 9.703)	4.881 (0.506; 9.256)
Baseline	Ref	Ref	Ref	Ref	Ref
Obese class I	11.722 (9.838; 13.606)	4.335 (3.916; 4.753)	10.108 (8.540; 11.676)	−5.467 (−8.488; −2.447)	−4.384 (−6.871; −1.898)
Obese class II	24.227 (22.154; 26.300)	9.080 (8.620; 9.541)	20.127 (18.405; 21.850)	−7.271 (−10.573; −3.969)	−5.058 (−7.779; −2.337)
Obese class III	46.566 (44.296; 48.836)	17.071 (16.567; 17.574)	33.954 (32.061; 35.847)	−8.127 (−11.720; −4.533)	−6.032 (−8.992; −3.071)
Pre-obese	Ref	Ref	Ref	Ref	Ref
Practice Nurse	−0.544 (−2.023; 0.935)	−0.457 (−0.779; −0.136)	−1.203 (−2.403; −0.002)	1.702 (−0.304; 3.708)	1.553 (−0.068; 3.174)
Other/Not Stated	3.987 (1.398; 6.576)	0.204 (−0.358; 0.767)	2.875 (0.774; 4.975)	2.814 (−0.669; 6.296)	1.505 (−1.307; 4.316)
General Practitioner	Ref	Ref	Ref	Ref	Ref
Retired	−1.631 (−3.620; 0.358)	−0.069 (−0.501; 0.363)	0.426 (−1.189; 2.040)	0.034 (−2.663; 2.730)	−0.184 (−2.365; 1.996)
Benefits	0.599 (−1.925; 3.124)	−0.093 (−0.642; 0.455)	2.346 (0.298; 4.393)	−2.527 (−5.962; 0.908)	−3.201 (−5.987; −0.415)
Other/Not Stated	−2.178 (−4.238; −0.117)	0.011 (−0.437; 0.459)	0.923 (−0.749; 2.595)	−0.146 (−2.945; 2.653)	0.058 (−2.208; 2.325)
Employed/FTE	Ref	Ref	Ref	Ref	Ref
Male	16.325 (14.883; 17.767)	0.077 (−0.237; 0.390)	8.580 (7.410; 9.751)	2.255 (0.301; 4.209)	2.372 (0.793; 3.952)
Female	Ref	Ref	Ref	Ref	Ref
Age	−0.121 (−0.188; −0.054)	−0.014 (−0.029; 0.001)	0.012 (−0.043; 0.066)	−0.209 (−0.300; −0.118)	−0.212 (−0.286; −0.138)
Quintile 1: Most Deprived (0–20%)	−0.633 (−2.956; 1.691)	−0.074 (−0.589; 0.441)	1.569 (−0.361; 3.499)	1.447 (−2.220; 5.114)	−1.216 (−4.230; 1.799)
Quintile 2: (21–40%)	0.309 (−1.791; 2.409)	0.132 (−0.334; 0.598)	1.597 (−0.150; 3.344)	1.984 (−1.374; 5.342)	0.555 (−2.213; 3.322)
Quintile 3: (41–60%)	0.774 (−1.328; 2.876)	−0.057 (−0.524; 0.410)	1.440 (−0.312; 3.193)	2.698 (−0.655; 6.052)	0.692 (−2.073; 3.457)
Quintile 4: (61–80%)	1.395 (−0.754; 3.545)	−0.077 (−0.555; 0.400)	1.082 (−0.713; 2.878)	3.336 (−0.106; 6.779)	1.264 (−1.574; 4.101)
Quintile 5: Least Deprived (+80%)	Ref	Ref	Ref	Ref	Ref
Constant	80.495 (76.266; 84.723)	29.087 (28.164; 30.010)	93.679 (90.230; 97.127)	30.782 (24.792; 36.772)	21.653 (16.789; 26.518)
**Interactions**	**Weight**	**BMI**	**Waist**	**Godin**	**Godin (HCS)**
Time = 2 × Obese class I	−0.402 (−1.060; 0.255)	−0.171 (−0.415; 0.073)	−0.440 (−1.349; 0.469)	−0.198 (−4.256; 3.861)	−0.548 (−3.945; 2.849)
Time = 2 × Obese class II	−0.855 (−1.573; −0.137)	−0.342 (−0.608; −0.075)	−0.697 (−1.687; 0.292)	0.098 (−4.358; 4.554)	−0.355 (−4.090; 3.380)
Time = 2 × Obese class III	−1.585 (−2.353; −0.817)	−0.630 (−0.915; −0.345)	−0.960 (−2.032; 0.111)	4.484 (−0.238; 9.205)	3.231 (−0.730; 7.193)
Time = 2 × Pre-obese	Ref	Ref	Ref	Ref	Ref
Time = 3 × Obese class I	−0.466 (−1.256; 0.325)	−0.187 (−0.516; 0.143)	−0.410 (−1.355; 0.535)	3.852 (−1.793; 9.497)	0.540 (−4.038; 5.118)
Time = 3 × Obese class II	−0.620 (−1.487; 0.248)	−0.237 (−0.598; 0.125)	−0.378 (−1.413; 0.658)	6.146 (0.060; 12.231)	2.824 (−2.132; 7.781)
Time = 3 × Obese class III	−2.316 (−3.242; −1.390)	−0.942 (−1.328; −0.556)	−0.920 (−2.035; 0.196)	10.234 (3.044; 17.424)	6.070 (0.152; 11.989)
Time = 3 × Pre-obese	Ref	Ref	Ref	Ref	Ref
Time = 2 × Quintile 1: (0–20%)	0.515 (−0.276; 1.305)	0.192 (−0.100; 0.485)	0.937 (−0.157; 2.031)	−6.321 (−11.108; −1.533)	−3.403 (−7.411; 0.605)
Time = 2 × Quintile 2: (21–40%)	0.551 (−0.169; 1.271)	0.260 (−0.007; 0.527)	0.274 (−0.722; 1.270)	−5.633 (−10.101; −1.165)	−2.858 (−6.590; 0.874)
Time = 2 × Quintile 3: (41–60%)	−0.090 (−0.815; 0.634)	0.052 (−0.217; 0.321)	−0.362 (−1.364; 0.640)	−1.940 (−6.475; 2.595)	−1.402 (−5.197; 2.393)
Time = 2 × Quintile 4: (61–80%)	0.423 (−0.326; 1.172)	0.206 (−0.073; 0.484)	0.235 (−0.802; 1.272)	−4.872 (−9.520; −0.223)	−3.941 (−7.831; −0.052)
Time = 2 × Quintile 5: (+80%)	Ref	Ref	Ref	Ref	Ref
Time = 3 × Quintile 1: (0–20%)	0.898 (−0.033; 1.830)	0.460 (0.071; 0.848)	0.923 (−0.192; 2.039)	−2.283 (−9.400; 4.835)	−0.107 (−5.943; 5.728)
Time = 3 × Quintile 2: (21–40%)	0.317 (−0.551; 1.185)	0.206 (−0.156; 0.567)	−0.111 (−1.150; 0.927)	5.711 (−0.591; 12.014)	5.561 (0.363; 10.758)
Time = 3 × Quintile 3: (41–60%)	−0.055 (−0.939; 0.829)	0.086 (−0.283; 0.455)	−0.910 (−1.971; 0.151)	2.607 (−3.705; 8.919)	5.327 (0.170; 10.485)
Time = 3 × Quintile 4: (61–80%)	0.208 (−0.699; 1.115)	0.240 (−0.139; 0.618)	−0.447 (−1.536; 0.643)	0.540 (−5.861; 6.942)	1.211 (−3.971; 6.392)
Time = 3 × Quintile 5: (+80%)	Ref	Ref	Ref	Ref	Ref

Confidence intervals in parentheses.

**Table 4 ijerph-17-05297-t004:** Qualitative study participant characteristics.

Participant Pseudonym	Age Group (Years)	Gender	Employment Status	Index of Multiple Deprivation Quintile	Number of Interviews Completed	Participation Status *	Self-Reported Weight Loss
Amy	20–29	Female	Home-maker	21–40% most deprived	2	Adherer	3.2 kg
Julie	50–59	Female	Employed	20% least deprived	2	Adherer	3.6 kg
Patricia	60–69	Female	Retired	40–60% most deprived	2	Adherer	3.2 kg
Paul	50–59	Male	Disability benefit	20% most deprived	2	Medically excluded	No weight loss
Jackie	40–49	Female	Carer	20% most deprived	2	Non-attender	No weight loss
Vicky	40–49	Female	Unemployed	20% most deprived	1	Dropout	Unknown
Louise	40–49	Female	Employed	20% most deprived	1	Dropout	Unknown

* Medically excluded: attended initial consultation but was excluded from scheme participation due to medical reasons (physiological measures above scheme acceptance guidelines e.g., blood pressure ≥180/100 mmHg or resting heart rate ≥100 beats per minute); Non-attender: attended initial consultation, but no exercise sessions; Dropout: attended initial consultation and some exercise sessions but informed study team they had stopped attending; Adherer: attended initial consultation and informed study team they were still attending exercise sessions.

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
