# Peer review of "The Northumberland Exercise Referral Scheme as a Universal Community Weight Management Programme: A Mixed Methods Exploration of Outcomes, Expectations and Experiences across a Social Gradient"

_ijerph, 2020, doi:10.3390/ijerph17155297_

Round 1

Reviewer 1 Report

Attached is a document with specific comments regarding the article. Several revisions are necessary, the greatest weakness of the article seems to me to be the section of materials and methods, wich is unclea, wich leads to some doubts in scientific terms.

Author Response

We have read carefully the feedback from all three reviewers and provide our responses point-by-point, in the following pages. We also highlight in yellow, associated changes to the manuscript. We consider that these changes have improved the manuscript and hope that you find them satisfactory.

Reviewer 1

Attached is a document with specific comments regarding the article. Several revisions are necessary, the greatest weakness of the article seems to me to be the section of materials and methods, which is unclear, which leads to some doubts in scientific terms.

We have made a number of revisions, detailed below, including to the materials and methods.

Scientific Manuscript Review - The Northumberland Exercise Referral Scheme as a universal community weight management programme: a mixed methods exploration of outcomes, expectations and experiences across a social gradient.

General comments:

I consider that the problem of the article is relevant because its evidence can bring gains

in health, especially for the population that benefits from ERS.

Thank you for this positive feedback. We respond to each specific comment in turn, below. Changes are highlighted in the main text, in yellow.

Specific comments:

  1. Writing

The writing, structure and organization of the manuscript is in accordance with the

guidelines of the IJERPH.

  1. Title

The title reflects the content and problem studied.

  1. Abstract

The abstract does not comply with IJERPH standards. Exceeds the number of words

allowed. The methodology used is not clearly described, namely the type of study, the

sample and the methods used.

Page 1. We have reduced the abstract to 200 words as per the journal requirement and adjusted content to clearly describe the type of study, sample and methods used.

  1. Keywords

The keywords are representative of the subject studied and exposed.

  1. Introduction

A state of the art is made in relation to the study. The purpose of the study is clear and

objective. The justification for the choice and importance of studying this topic was not

presented.

Page 2 lines 55-62: We have amended elements of the Introduction section to emphasise the importance of this topic, as an area of research – and the associated gaps in the evidence-base. We also draw the reviewer’s attention to the penultimate paragraph of the introduction which states the importance of exercise referral (as a topic) in the wider physical activity landscape, particularly in respect of inequalities. We hope that this is now satisfactory.

  1. Materials and Methods

The methodology is confused, it is a mixed methodology (quantitative and qualitative

study), but the type of study is not mentioned at any time, as well as the choice of the

population and sample selection is not described. The quantitative data collection

instruments are adequate, but in relation to qualitative data, it is not clear how the

interviews were analyzed, which method was used, how the categories of responses

arrived. I suggest a more objective description of the methodology.

Page 3 lines 99-102: We have clarified the type of study at the beginning of the methods section by adding the statement:

The study employed an embedded mixed methods design [40], where the qualitative dataset provided a supportive secondary role to the quantitative data set. It encompassed a main, primary quantitative phase of data collection and a subsequent qualitative phase, implemented partway through quantitative data collection.

We have added in two further sections to describe:

  1. The quantitative sample (page 3, lines 122-128):

Anonymised data for all those referred to the Northumberland ERS between June 2009 and March 2014 with a primary reason for referral indicated as overweight or obesity (BMI>24.9kg/m2) were included in the quantitative element of the study (n=3,624). All those referred during this period with any other primary reason for referral were excluded from the study. Some predictive personal and referral characteristics have been previously reported for an initial wave of participants (n= 2233) who started the scheme between 2009 and 2010, but this included all referral conditions [12].

  1. The qualitative sample (page 4, lines 152-157)

We used early convenience sampling of referrals made with a primary referral reason of overweight or obesity during May-June 2013 to two of the nine leisure centres were eligible to take part in two semi-structured interviews (n=42). Referrals to these centres included a broad adult age range, males and females, and a range of economic circumstances. Later recruitment used purposive, heterogeneity sampling [45] to target those <50 years old.

Page 5, lines 207-219. We have added detail to the qualitative analysis section for further clarity. We already state that we use the framework approach to thematic analysis.

Interviews took place between May and December 2013, were audio recorded and transcribed verbatim. All data including references to weight were extracted into a Microsoft Excel spreadsheet. Pseudonyms were used and all data were thematically analysed using the framework approach [53]. CLH and CDR familiarized themselves with transcripts through several readings and CH checked accuracy against audio recordings. We separately used line by line coding to openly record preliminary concepts and patterns for both interviews for two participants and ordered these using Microsoft Excel. using manual processes. This resulted in 18 initial codes including personal perceptions of weight, perceptions of current diet, personal support for weight loss, weight loss expectations, other expectations from scheme participation, age considerations, benefits of participation, and support from primary care. After discussion, we established agreed codes to form an initial analytical framework of four themes; support in primary care, personal circumstances, expectations versus rewards, and wellbeing strategies. We analysed one more set of transcripts, before refining and finalising of the analysis framework, condensing the initial framework to three final themes. We mapped and explored connections within and between participants and categories using a matrix. Our analysis developed, reviewed and finalised themes that identified the complexity of, and possible explanations for, the ERS experiences of those referred for weight loss.

  1. Results

The displayed results are clearly organized; and tables and figures help your

interpretation.

  1. Discussion

The discussion should present the results in context with the existing literature, it

presents some confrontation, but I believe that the results should be compared with

more studies. The implications of the study for science are referred to and suggest

further investigations.

Unfortunately, the ERS data available for weight-related referrals is scarce, however we have attended to this in the opening paragraph of new section 4.1. In this section we have now also placed our findings in the context of other largescale RCT-based and observational studies (wider referral pool) which have shown improvements in physical activity (Campbell et al., 2015; Murphy et al., 2012; Rowley et al., 2019). We additionally note the challenges of direct comparison due to differing PA data collection methods, ERS duration and referral reasons. Hopefully this now presents a more contextualised narrative discussion. (Discussion, lines 448-467).

  1. Conclusion

The main conclusions of the study are mentioned here. The authors respond to the

objectives set for the study. Just as they value the contributions of the work in relation

to the area in question.

  1. References

The authors made an effort to use current references, most of which date from the last

5 years. Almost all of them need to be reviewed, as they do not comply with the IJERPH

guidelines, especially with regard to the short name of the magazines.

Apologies, this was an oversight at the point of submission when our Endnote list was not properly reformatted. We have attended to this and the list should now comply with the journal guidelines.

Reviewer 2 Report

Thank you for giving me the opportunity to revise this manuscript. I consider that is an article very completely and interesting because they explore the impact on weight status and PA of an exercise referral schemes, which it is very interesting.

I consider that the authors offer a very complete information of their study with a lot of data. I only have some recommendations for improve this article:

  • In line 173, you explain the BMI classification according the WHO. Please, include the reference of this classification!
  • The use of convenience sampling for qualitative procedure is missing as a limitation in the limitation section.
  • It would be interesting that the authors explain better the implications of this study for clinical practice.
  • In conclusion section, might you identify future directions?

Author Response

Responses to Reviewer 2

We have read carefully the feedback from all three reviewers and provide our responses point-by-point, in the following pages. We also highlight in yellow, associated changes to the manuscript. We consider that these changes have improved the manuscript and hope that you find them satisfactory.

Thank you for giving me the opportunity to revise this manuscript. I consider that is an article very completely and interesting because they explore the impact on weight status and PA of an exercise referral schemes, which it is very interesting.

Thank you for these constructive comments. We respond to each in turn, below. Changes are highlighted in the main text, in yellow.

I consider that the authors offer a very complete information of their study with a lot of data. I only have some recommendations for improve this article:

  • In line 173, you explain the BMI classification according the WHO. Please, include the reference of this classification!

We have included a reference as requested on page 4, line 174.

  • The use of convenience sampling for qualitative procedure is missing as a limitation in the limitation section.

We have added in the following to the limitations section of the manuscript (Discussion, lines 612-614),

‘We used early convenience sampling to recruit participants due to ERS staff facilitating recruitment, and note previous criticism of this strategy [68, 69]. However, we combined this with purposive sampling after the first three participants, to include those under 50 years.’

  • It would be interesting that the authors explain better the implications of this study for clinical practice.

We have now explicitly highlighted implications for clinical practice in the discussion and conclusions section. (see lines 591; 599-600; 622-628).

  • In conclusion section, might you identify future directions?
    We have now added some information to the conclusion, to clearly identify future directions (lines 625-628).

Reviewer 3 Report

This manuscript describes an evaluation of an exercise referral scheme to address obesity. The sample is large, and the study is potentially of interest.

There are two main aims: i) Examine weight loss and PA change across social determinants of health from record review; ii) Understand the expectations and experiences of participants in the ERS from interviews with sub-group. These aims are complementary.

Although I did not find anything wrong with this manuscript the discussion is very dense and could perhaps be shortened and focussed a bit more to make the messages from this study clearer.

Author Response

Responses to Reviewer 3

We have read carefully the feedback from all three reviewers and provide our responses point-by-point, in the following pages. We also highlight in yellow, associated changes to the manuscript. We consider that these changes have improved the manuscript and hope that you find them satisfactory.

This manuscript describes an evaluation of an exercise referral scheme to address obesity. The sample is large, and the study is potentially of interest.

There are two main aims: i) Examine weight loss and PA change across social determinants of health from record review; ii) Understand the expectations and experiences of participants in the ERS from interviews with sub-group. These aims are complementary.

  • Although I did not find anything wrong with this manuscript the discussion is very dense and could perhaps be shortened and focussed a bit more to make the messages from this study clearer

Thank you for these constructive comments. We have made some changes throughout the discussion, as well as the structure, in order to reduce density of the text. We have also added in subheadings to clarify key messages (Discussion line 448 onwards). Changes are highlighted in the main text, in yellow.

Round 2

Reviewer 1 Report

The authors made an effort to review all points of the article that had some limitations, wich resulted in na excelente final version of the article. The presentation of the methodology was very clear and objective. I believe that the article is ready to be published.